# Spatial and temporal dynamics of leptospirosis in South Brazil: A forecasting and nonlinear regression analysis

**Alessandra Jacomelli Teles[1], Bianca Conrad Bohm[2], Suellen Caroline Matos Silva[2], Nádia Campos Pereira Bruhn[3], Fábio Raphael Pascoti Bruhn[4]***

**1** Secretaria Municipal de Saúde de Herval, Herval, Rio Grande do Sul, Brasil, **2** Universidade Federal de Pelotas, Programa de Pós Graduação em Veterinária, Capão do Leão, Rio Grande do Sul, Brasil, **3** Centro de Integração do Mercosul, Universidade Federal de Pelotas, Pelotas, Rio Grande do Sul, Brasil, **4** Departamento de Veterinária Preventiva, Universidade Federal de Pelotas, Capão do Leão, Rio Grande do Sul, Brasil

* fabio_rpb@yahoo.com.br

**Data Availability Statement:** All relevant data are within the manuscript and its Supporting Information files.

## Abstract

Although leptospirosis is endemic in most Brazilian regions, South Brazil shows the highest morbidity and mortality rates in the country. The present study aimed to analyze the spatial and temporal dynamics of leptospirosis cases in South Brazil to identify the temporal trends and high-risk areas for transmission and to propose a model to predict the disease incidence. An ecological study of leptospirosis cases in the 497 municipalities of the state of Rio Grande do Sul, Brazil, was conducted from 2007 to 2019. The spatial distribution of disease incidence in southern Rio Grande do Sul municipalities was evaluated, and a high incidence of the disease was identified using the hotspot density technique. The trend of leptospirosis over the study period was evaluated by time series analyses using a generalized additive model and a seasonal autoregressive integrated moving average model to predict its future incidence. The highest incidence was recorded in the Centro Oriental Rio Grandense and metropolitan of Porto Alegre mesoregions, which were also identified as clusters with a high incidence and high risk of contagion. The analysis of the incidence temporal series identified peaks in the years 2011, 2014, and 2019. The SARIMA model predicted a decline in incidence in the first half of 2020, followed by an increase in the second half. Thus, the developed model proved to be adequate for predicting leptospirosis incidence and can be used as a tool for epidemiological analyses and healthcare services. Temporal and spatial clustering of leptospirosis cases highlights the demand for intersectorial surveillance and community control policies, with a focus on reducing the disparity among municipalities in Brazil.

## Author summary

The southern region of Brazil has the highest morbidity and mortality from leptospirosis in the country. Here, we present an approach based on spatial and temporal modeling to help understand the incidence of leptospirosis in Rio Grande do Sul, an endemic state located in southern Brazil. Clusters of disease incidence and mortality were observed in

**Funding:** This study was funded by the Fundação de Amparo a Pesquisa do Rio Grande do Sul (FAPERGS) (funding code 21/2551– 0000608–0 to FRPB) and Conselho Nacional de Desenvolvimento Científico e Tecnológico (CNPq) (funding code 316426/2021- 0 to FRPB) and Coordination for the Improvement of Higher Education Personnel (CAPES) (funding code 001 to BCB). The founders had no role in study design, data collection and analysis, publication decision, or manuscript preparation.

**Competing interests:** The authors have declared that no competing interests exist.

Centro Oriental Rio Grandense and Metropolitan of Porto Alegre mesoregions, while no cases were recorded in 220 municipalities between 2007 and 2019. A model with satisfactory predictive ability was also developed. The high underreporting of cases may reflect the failure in the sensitivity of the state's leptospirosis surveillance system. These results can assist public healthcare services in allocating appropriate efforts and resources to control the disease, specifically in these regions with a higher risk of leptospirosis, considering the growing need to clarify the dynamics of this neglected disease in Brazil.

## Introduction

The febrile illness leptospirosis is globally one of the most widespread emerging zoonoses [1]. It has a high transmission capacity among socio-environmental vulnerable populations [2,3,4]. Hence, this disease is a relevant public health concern in Brazil, with social, health, and economic impact; high hospital cost; loss of productivity; and high lethality [5,6]. Leptospirosis is caused by *Leptospira interrogans*, a bacterial species that survives in varied environments for prolonged periods and affects humans and wild, domestic, and synanthropic animals, which then become carriers of this species and contribute to its spread in nature [7].

In Brazil, leptospirosis is an endemic disease and is present throughout the country. It has a high incidence, with an annual average of 3,926 confirmed cases and a death rate of 8.9% between 2007 and 2016, with the highest number of cases registered in the southeast and south regions of Brazil [8,9,10]

Leptospirosis is easily ignored, and relatively little is known about it, as few studies have been conducted on this disease; hence, it is considered a neglected disease. In this context, reliable data on the incidence and prevalence of leptospirosis in different areas are still scarce [11]. Therefore, we consider that the analysis of real-time data on the incidence of confirmed leptospirosis cases collected from information systems derived from the official surveillance of the disease in Brazil can contribute to elucidate the difficulties in disease control and to understand the dynamics of occurrence of this endemic disease in the country. These observations can be used as tools to help healthcare managers to design appropriate prevention strategies and promote public health safety efforts [12,13].

The surveillance data of transmissible diseases are useful to healthcare managers to monitor daily incidence trends, and the usefulness of these data can be further extended by subjecting them to appropriate statistical analyses [14]. This approach can be used to find gaps in the notification system and to identify the disease incidence peak periods in a season [15]. In addition to these benefits, predictive analytical methods provide the scope to predict the future burden of the disease and to early predict an epidemic [16,17].

The present study aimed to analyze the spatial and temporal dynamics of leptospirosis incidence and lethality rates during 2007–2019 at the municipal and state levels in Rio Grande do Sul, Brazil, and to predict the expected incidence of the disease in the subsequent period. On the basis of these results, we expect to contribute to the development of more effective strategies to control leptospirosis in high incidence regions such as southern Brazil.

## Methods

### Ethics statement

The study was approved by the Ethics Committee of the Faculdade de Medicina of Universidade Federal de Pelotas (Approval No.: CAAE 46714421.0.0000.5317) in accordance with all

ethical principles and current legislation for research involving human beings. Data confidentiality was thus ensured, and the data were used only for research purposes.

## Study design

An ecological study was conducted using data collected during the routine surveillance of leptospirosis in the state of Rio Grande do Sul between 2007 and 2019. The dynamics of this disease in the state were analyzed through spatiotemporal statistical analyses. An incidence prediction model for the subsequent year was then developed.

## Study location

Rio Grande do Sul is the southern most state in Brazil, and it is divided into 497 municipalities (Fig 1). It has an estimated population of 11,422,973 inhabitants in 2020 [18] and an area of 281,730.149 km$^2$, with a population density of 39.79 inhabitants/km$^2$. It is located at latitude 27°03′42″ and 33°45′09″ South and longitude 49°42′41″ and 57°40′57″ West. The climate is subtropical temperate, with a large seasonal variation in temperature. The average temperature ranges between 15°C and 18°C, with a minimum of up to –10°C and a maximum of 40°C. The rainfall volume varies among the regions, with an average annual precipitation of 1,500 mm. The state receives a relatively balanced distribution of rainfall throughout the year [19].

The literacy rate is >95%. The gross domestic product *per capita*, according to the statistical data from 2017, is close to R$37,000/year. In 2015, 93.7% of the population had access to a direct or indirect garbage collection process, and 40% of people had the facility of sewage collection networks around their homes [18].

There are three major economic regions: (1) the south region with a larger concentration of land, large livestock farms, and mechanized planting of rice, soybean, and wheat. This area also has greater income inequality; (2) the northeast region that includes the state capital, with

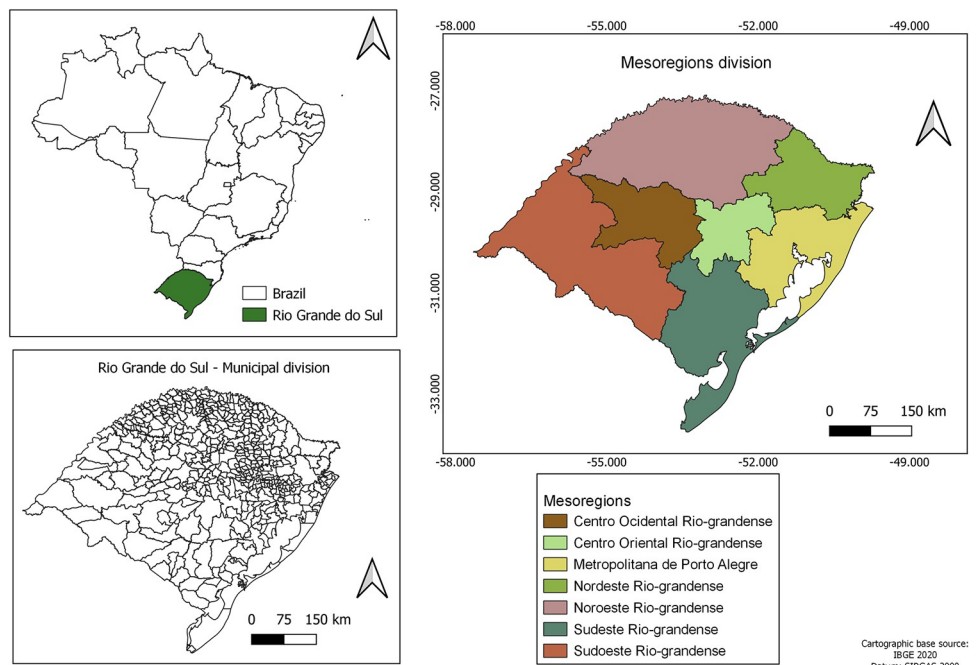

**Fig 1. Location of the Rio Grande do Sulstate in Brazil, with its divisions according to mesoregions.**

more industries and predominantly small properties; and 3) the northern region, mostly colonized by European immigrants, with a greater forest cover, valleys, and plains with small agricultural lands [19](Fig 1).

## Data collection

Since 2000, it became mandatory to notify human leptospirosis cases through the Ministry of Health's Information System for Notifiable Diseases (acronym in Portuguese: SINAN) by using a passive surveillance system. In 2006, the notification form was updated to include minor modifications, and the updated version was implemented across the country in 2007 [10]. Information on the number of confirmed cases of leptospirosis in Rio Grande do Sul from 2007 to 2019, as reported in the SINAN, was obtained through the Datasus system, Tabnet, from *Ministério da Saúde* [5,20]. The cases included in this study were confirmed on the basis of clinical-epidemiological or laboratory criteria in accordance with the Epidemiological Surveillance Guide of the Ministry of Health [21].

## Statistical analysis

An ecological study of time series and spatial analysis was performed, which enabled identifying geographical and temporal patterns of the disease in Rio Grande do Sul. By using leptospirosis data collected over 13 years of the observation period, incidence rates (per 100,000 inhabitants) and lethality rate (%) were calculated using the population projection of Instituto Brasileiro de Geografia e Estatística for each year of interest [21].

These indicators were annually mapped in all municipalities in Rio Grande do Sul to confirm the existence of clusters and thus analyze the disease risk in different regions of the state. The dynamics of the incidence rates at the state level between 2007 and 2019 were analyzed by a generalized additive model (GAM), while the number of cases for 2020 for Rio Grande do Sul was forecasted using a seasonal autoregressive integrated moving average (SARIMA) model.

## Spatial clusters

Hotspot analysis evaluates the distribution of events in a given area. It enables one to visually identify the concentration of an event and indicate the concentration areas where the phenomenon occurs frequently. This analysis was performed to detect spatial clusters with a high concentration of high incidence of leptospirosis, called hotspots, between 2007 and 2019, by using the Getis-Ord Gi statistic [22].

The hotspots are taken as central points, where there is a greater intensity of the analyzed event in the study area. By considering the circular distance of each central point, it scores all points within the sample influence radius with event smoothing [9,23]. "Heat maps" were constructed to visualize the results of hotspot analysis with the highest leptospirosis incidence areas [24].

Spatial statistical analyses were performed using the Geographic Information System of Quantum GIS software (QGIS), version 3.4.7.

## GAM

GAM represents an extension of the generalized linear model [25] and an alternative for modeling nonlinear relationships with an undefined shape. This model is based on nonparametric functions known as smoothing curves, in which the association shape is defined by the data [26,27].

The general formula of GAM with Poisson likelihood was as follows:

$$y_i \sim Poisson\,(Z_i)$$

$$\log(Z_i) = b_0 + \sum s_j(x_{ij}k),$$

where $y_i$ is the observation $i$, $Z_i$ is the linear predictor for the observation $i$, $b_0$ is the intercept, $s_j$ is the spline for the predictor $x_{ij}$, and $k$ is the number of knots [28]

GAM allows a wide range of distribution for the adopted response variable as well as linkage functions to measure the effects of predictor variables on the dependent regressors [25,29]. In the present study, a smoothing function was used in the year variable to confirm the dependence of observations on time. The response variable was the number of confirmed leptospirosis cases in year $i$ with Poisson distribution, with the population as the offset term and a spline function in the continuous time variable [30]. The suitability of the data for Gaussian and negative binomial distribution was tested (S1 Table and S1 Fig); however, because the response variable is a count, the Poisson distribution with a log link was used [30,31].

Smooth terms were specified to select the GAM by using the mgcv package in R studio software. We considered the following smoothing terms to select the best model fit: (i) thin plate regression splines; (ii) Duchon splines; (iii) cubic regression splines; (iv) B-splines; and (v) P-splines. The knot-based penalized cubic regression splines showed the best performance. The unbiased risk estimator (UBRE) is essentially scaled according to Akaike's information criterion (AIC) (generalized case) [32]. UBRE and the percentage of deviance explained were the criteria used to identify appropriate smoothness and select the best model fit (S1 Table and S1 Fig).

The analysis was performed using R packages mgcv and ggplot2 [33].

## SARIMA model

Time series analysis was used to describe the leptospirosis trend between 2007 and 2019 and to forecast the disease incidence in 2020. The components of time series analyses were trend element evaluation, seasonality, cyclical variation, association, and random variation. Rio Grande do Sul experiences hotter months from November to April and colder months from May to October [19,34]. In the present study, seasonality was tested to detect whether leptospirosis occurrence was concentrated in some of these months. This assessment enabled us to understand the implicit processes over time and thus provide important information for planning public health policies for preventing the disease [35].

In the present study, for time series analysis, trend and seasonality components were evaluated together with SARIMA model adjustment.

Time series can be decomposed into the following equation:

$$Y_t = T_t + S_t + a_t$$

where trend ($T_t$) refers to an increasing or decreasing pattern during the observation period, seasonality ($S_t$) shows fluctuations occurring in periods shorter than one year, and the random component or error ($a_t$) shows irregular random oscillations [36]. Several random factors, such as climate and health policy changes, also need to be considered while evaluating leptospirosis incidence through a monthly series. These factors are known as white noise, as they are random elements that occur due to probability and are not related to cyclical variations or trends.

The SARIMA time series model was used to assess trend and seasonality components and their adjustment [36]. Cases from 2007 to 2019 were used to develop the logical system. Data

from the year 2019 were used for validation and to predict the incidence in the last year of the study, i.e.,2020. To validate the adopted model choice, the real values were compared with the predicted ones, and their 95% prediction interval was calculated. Mean absolute percent error (MAPE) values and Theil's U statistic were determined. Values less than 1 indicated the adequacy of the forecast [37,38].

Monthly incidences were transformed by adding the first difference to the logarithmic variable to form a stationary series. In the months in which leptospirosis occurrence was 0 (zero), one unit was assigned. The least squares method was used to confirm trends and seasonality (p<0.05). The augmented Dickey–Fuller test (ADF) was performed to test data stationarity (p<0.05) [39].

The adequacy of each model was confirmed using the residual autocorrelation function (ACF) and partial autocorrelation function (PACF) histogram graphs; furthermore, the Ljung–Box test was performed to investigate the randomness of the residuals. To compare adjustments for different models, the AIC and Hannan-Quinn and Schwarz alternative models were used [38–40] (S2 Table). Time series analyses were performed using Gretl 1.9.12 software [41].

## Results

From 2007 to 2019, 6,162 cases of human leptospirosis and 321 deaths due to the disease were confirmed in Rio Grande do Sul. The average incidence rate was 4 cases per 100,000 inhabitants, and the lethality rate was 5%. Fig 2 shows the spatial distribution of leptospirosis incidence and lethality rates. In several regions of the state, high leptospirosis incidence rates were observed, particularly in Metropolitana de Porto Alegre, Centro Oriental Rio Grandense, and in recent years, in the Noroeste Rio-Grandense mesoregion. In addition, 220 municipalities did not confirm any case in the entire study period, while deaths from leptospirosis were recorded in 59 municipalities, with the highest number of deaths recorded in the capital, Porto Alegre (60 deaths, lethality rate: 8.9%).

Fig 3 shows annual maps of the areas with the highest concentration of high leptospirosis incidence, represented by density levels according to shade. The hotspots, i.e., central points represented in red, demonstrate a high concentration of high leptospirosis incidence rates. Around these hotspots, the tonality softens (yellow color), indicating a decrease in the incidence rate. The mapping shows hotspots in the Porto Alegre Metropolitan and Centro Oriental Rio Grandense mesoregions in all years. Thus, possibly in these places, there is a high risk of leptospirosis transmission.

By using GAM, a leptospirosis nonlinear regression model was developed for the cases over the years (Fig 4A). An increase in the disease occurrence risk was observed in Rio Grande do Sul from 2010, with a peak in 2011, followed by a decrease in risk until 2012. Between 2013 and 2019, the disease risk annually increased, with new peaks in the years 2014 and particularly 2019. Similarly, in the analyzed leptospirosis incidence time series between 2007 and 2019, incidence peaks were observed in 2011 and 2019. Furthermore, time series analysis revealed that the disease was seasonal in the months of January, February, March, and April (p<0.05) (Fig 4B). After differentiating the time series, the necessary stationarity for the SARIMA model was confirmed using the ADF test, in addition to the ACF and PACF (Fig 4B).

Among the leptospirosis incidence time series models, SARIMA (1,2,1) (1,0,1) was the best model. For confirming the model adequacy, the Theil-U statistic value was 0.683, and the MAPE value was 50.0%, thus, indicating acceptable forecast errors; this suggested that the proposed model was adequate to predict future leptospirosis incidence. After adjusting the model, the original series data (red line) and the predicted values (blue line) in the period under analysis were compared, which allowed leptospirosis incidence prediction in Rio Grande do Sul for

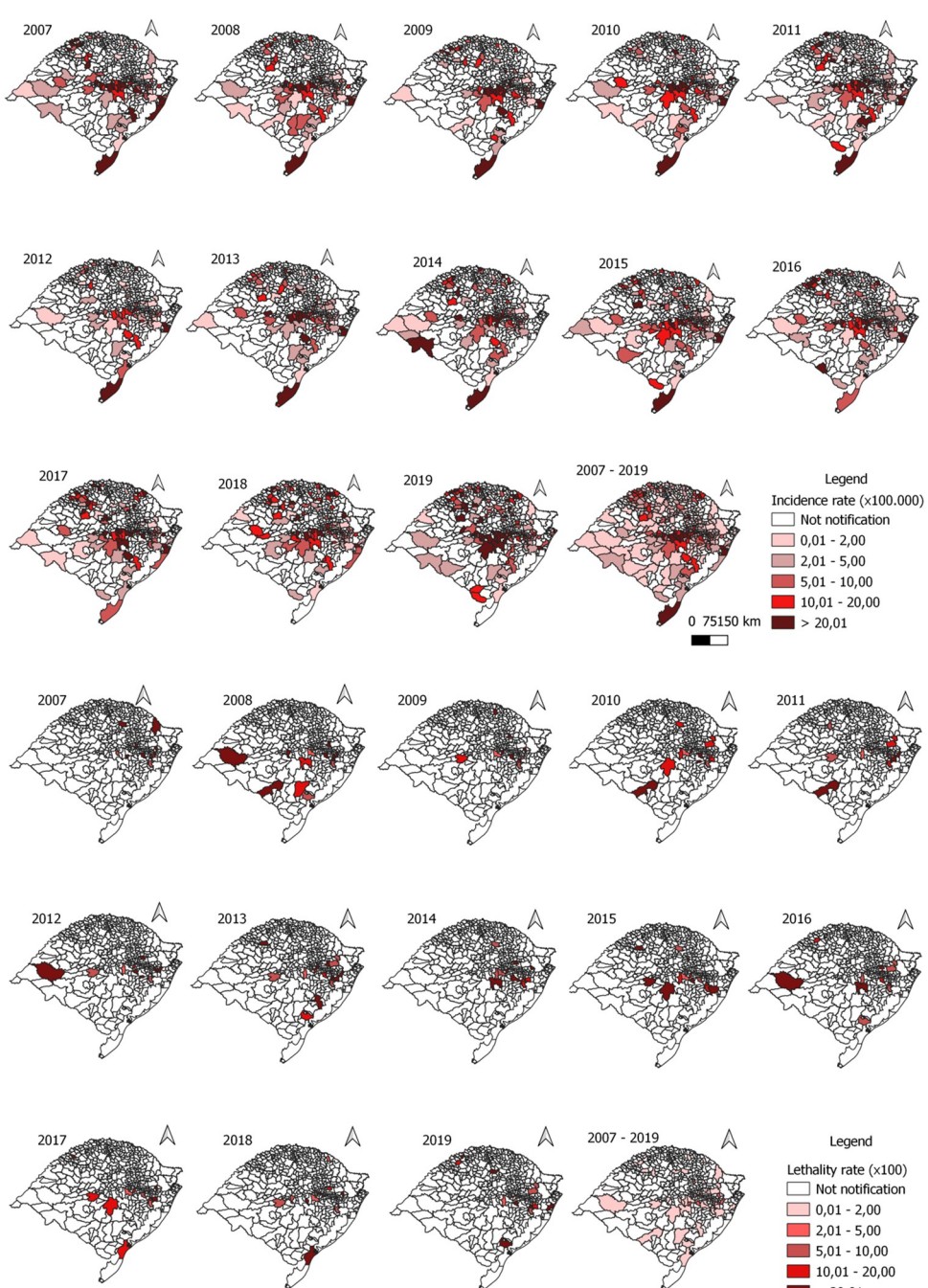

**Fig 2. Spatial distribution of leptospirosis incidence and lethality rate in Rio Grande do Sul municipalities, Brazil, from 2007 to 2019 (incidence rate: cases/100,000 inhabitants; lethality rate: %).**

the year 2020 and the prediction interval (black bars) (Fig 5). The model considered the 12-month seasonal pattern and predicted a decrease in incidence in 2020 (Fig 5).

## Discussion

Our present study demonstrated the heterogeneity of the endemic occurrence of leptospirosis in Rio Grande do Sul, Brazil. The studied time-series highlighted the difficulties in controlling

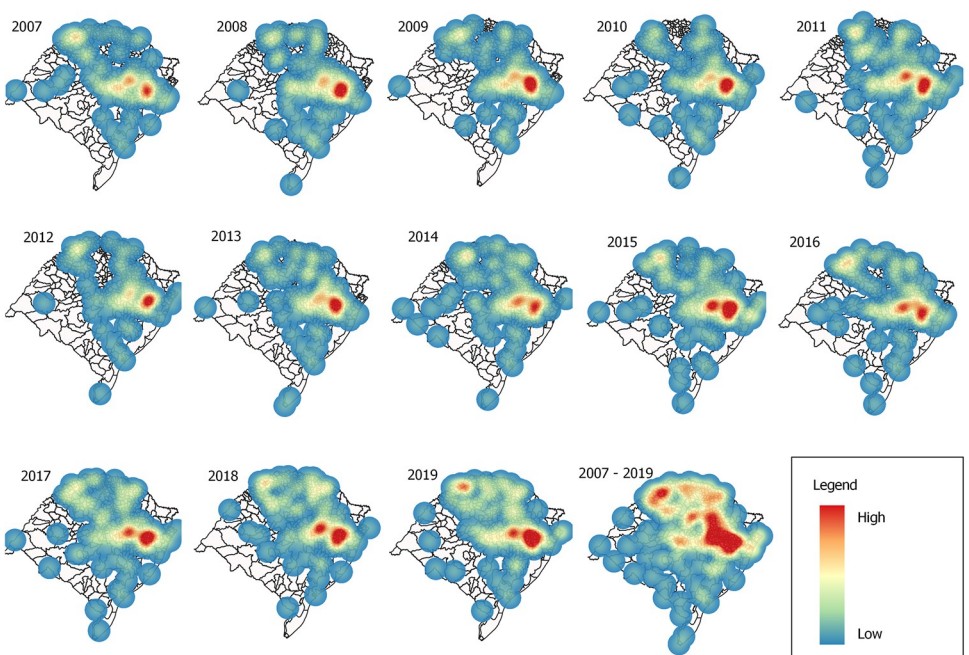

**Fig 3. Hotspot analysis of leptospirosis incidence from 2007 to 2019 in Rio Grande do Sul, Brazil.**

this disease, given the increase in the number of leptospirosis cases in the last years of the evaluated time series. It is important to highlight that leptospirosis is a neglected disease distributed in all Brazilian regions, with a higher prevalence in south and southeast states [9,10]. The disease spreads in Brazil, particularly in the urban environment, because of the high density of

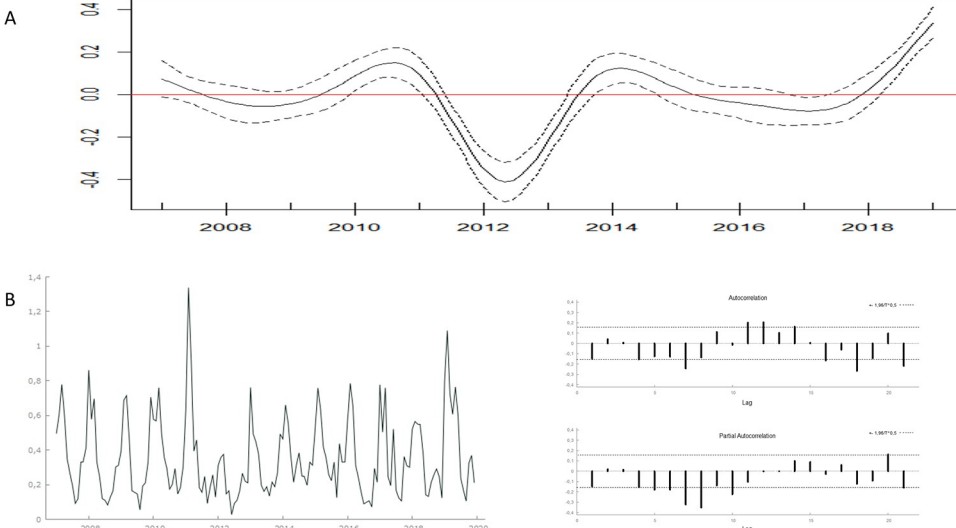

**Fig 4.** Temporal distribution of leptospirosis incidence rate (per 100,000 inhabitants) between 2007 and 2019 in Rio Grande do Sul, Brazil, (A) estimated by Generalized Additive Model (GAM) with Poisson distribution and a spline function in the continuous time variable, with 95% confidence interval ($p<0.01$, $R^2 = 75.9\%$) and (B) time series descriptive evaluation in the period, with the autocorrelation function (ACF) and the partial autocorrelation function (PACF) correlograms.

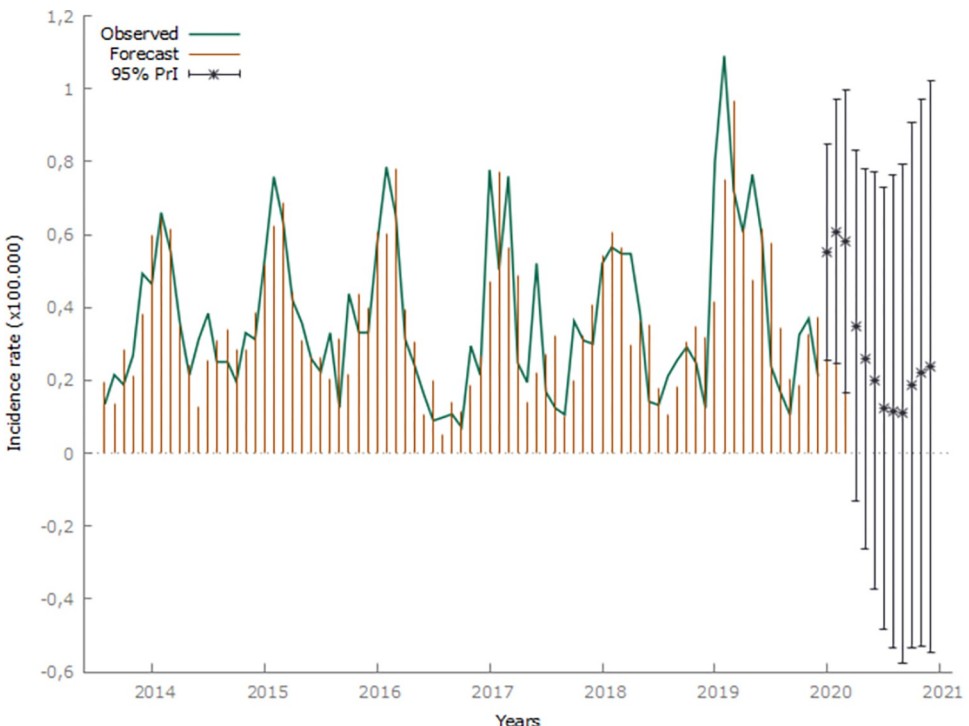

**Fig 5. Monthly leptospirosis incidence time series distribution in Rio Grande do Sulbased on data from 2007 to 2019 and predicted values for 2020, with a 95% prediction interval (95% PrI), after Seasonal Autoregressive Integrated Moving Average (SARIMA) model adjustment.**

hosts, uncontrolled population growth, and the presence of large areas of poverty in the country.

In Brazil, the epidemiological data of leptospirosis may be nonexistent and inaccurate [42], given that the often nonspecific or silent clinical presentation of the disease contributes to its underreporting [43]. Still, it is much more difficult to raise awareness about the need to notify the suspected cases of leptospirosis by health professionals [6,44]. In this context, it was observed that the distribution of leptospirosis incidence rates in Rio Grande do Sul varied greatly in the different regions of the state; in 20 municipalities, it was very high in the evaluated period, while in 220 municipalities, cases the disease was not recorded in the years evaluated. This finding may reflect the failure in the sensitivity of the state's leptospirosis surveillance system.

Nevertheless, in 2019, the incidence rate of leptospirosis in Brazil was 1.73 cases/100,000 inhabitants, while in Rio Grande do Sul, it was almost four times higher, reaching 6.1 cases/ 100,000 inhabitants [45]. Hotspot analysis proved to be useful for identifying clusters of leptospirosis, as this analysis identified clusters located in the mesoregions with the highest incidence rates in the state: Metropolitan of Porto Alegre (4.08 cases/100,000 inhabitants) and Centro Oriental Rio Grandense (16.7 cases/100,000 inhabitants). The municipalities in these mesoregions have the highest population density, and a smaller territorial extension in Rio Grande do Sul [46]. The Metropolitan of Porto Alegre is the most urbanized mesoregion (93.2%), while the Centro Oriental Rio Grandense is the least urbanized mesoregion (69.8%) in the state [46]. These characteristics indicate that these mesoregions have different profiles of disease occurrence, predominantly urban or rural, and constitute the two highest risk areas for human leptospirosis in the state in all years between 2007 and 2019. By using different

approaches, other authors have already confirmed that the same regions of Rio Grande do Sul are high-risk clusters for the disease [10,47]; this finding demonstrates the maintenance of incidence clusters in the state over the last decades.

With regard to deaths due to the disease, between 2007 and 2019, the average leptospirosis lethality was 5.3% in Rio Grande do Sul, while it was 8.6% (56.6% higher) in the Metropolitan of Porto Alegre mesoregion. The lethality rate was lower at 4.0% in the Centro Oriental Rio Grandense mesoregion. On the other hand, this mesoregion had the highest mortality rates due to leptospirosis from 2007 to 2019, with an average of 0.61 deaths/100,000 inhabitants, which was higher than the average rates for the entire country (0.17 deaths/100,000 inhabitants) [45], Rio Grande do Sul (0.23 deaths/100,000 inhabitants), and the Metropolitan of Porto Alegre mesoregion (0.32 deaths/100,000 inhabitants). These high morbidity and mortality indicators demonstrate the severity of leptospirosis in Rio Grande do Sul and reinforce the importance of allocating efforts to control this disease in the state, particularly in the regions of the clusters identified in the present study in all years between 2007 and 2019.

In urban areas, the greatest risk of leptospirosis transmission occurs in areas with poor sanitation infrastructure, precarious housing, and prone to natural flooding due to rains [23]. The Metropolitan of Porto Alegre mesoregion, which constitutes a cluster of cases and deaths in the last decade, concentrates 38.2% of the total population of the state [19]; it is a low-altitude area, where cities often experience periodic flooding, which can facilitate disease transmission in the form of outbreaks [48].

In southern Brazil, leptospirosis incidence in rural areas is twice as high as that in urban areas [10]. Approximately 50% of the municipalities in Rio Grande do Sul are considered at risk for the disease, most of which are critical areas for leptospirosis [49]. The concentration of rural leptospirosis cases in the state is reported in the central region, mainly in the Centro Oriental Rio Grandense mesoregion, which produces tobacco, and the Sudeste Rio-Grandense, which produces rice [49], with a higher incidence in males [10]. Previous studies have also reported a higher incidence of the disease in the rural regions, with risk for workers, especially in places where synanthropic animals are kept as well as in places with exposure to production animals that may be infected [10]. Even though efforts have been made for worker health surveillance, there is much difficulty in adapting workers from rural segments to accept the use of personal protective equipment. Preventive strategies are increasingly necessary for this sector and must be intensified through educational campaigns.

In the present study, leptospirosis seasonality was observed from January to April, with a higher incidence in the hottest months. By performing SARIMA time series analysis, Warnasekara*et al*. [50] observed a higher leptospirosis incidence in wetlands in Sri Lanka, which also has higher rainfall and a higher number of rainy days per month. They attribute this high incidence to the difficulty of *Leptospira* sp. to survive for long periods in low temperatures, high altitudes, and high solar radiation. In Rio de Janeiro, from 2007 to 2012, Guimarães*et al*.[51] observed leptospirosis seasonality, with a higher concentration in the summer and an increasing trend in the number of cases in the period 2008–2010 because of an increase in rain precipitation. On the basis of time series analysis, they suggested a time interval between the precipitation peak and the appearance of symptoms, wherein the number of cases begins to increase approximately a month after the rainy season. This information is crucial to plan preventive or assistance strategies for the exposed communities.

The knowledge of temporal dynamics and the prediction of infectious disease outbreaks by using time series, particularly GAM and SARIMA, are the goals of several researchers [12,37]. GAM is often used in association analysis that does not show a linear pattern. This usually occurs with the temporal evolution of infectious diseases. In GAM, which was used to evaluate the temporal dynamics between 2007 and 2019, a smoothing function was used in the year

variable to confirm the dependence of observations on time to recognize the functional trend of the "year" variable [30]. Thus, it was possible to observe growth trends in the incidence rate, mainly from 2016 to 2019, in addition to peaks noted in the years 2010, 2014, and 2019. The SARIMA model used to predict cases enabled us to observe a perspective of an increase in cases, especially in the second half of 2020.

The estimation of the occurrence of future cases, based on predictive models such as SARIMA (1,2,1) (1,0,1), which offer an acceptable adjustment, can be highlighted as one of the approaches available to face the challenges related to the surveillance of leptospirosis and other infectious diseases in Brazil. The autoregression and moving average parameters showed that the number of leptospirosis cases in future periods could be estimated based on the case numbers in the previous months. The proximity of actual leptospirosis cases that occurred in 2019 (5.9 cases/100,000 inhabitants) and the SARIMA predicted number (3.9 cases/100,000 inhabitants; 95% CI = 1.9–6.7) showed that the model could be used to predict leptospirosis cases in the state. Case prediction as an instrument for planning control methods at the municipal level, associated with other determining variables and preventive measures, can highlight priorities in small areas of the urban territory and indicate the adoption of integrated strategies [12–52].

The use of secondary data from surveillance systems entails working with several limitations. We used secondary data where cases were reported by health professionals, healthcare services, and the public. Therefore, this case reporting was subject to misclassification bias. The municipal epidemiological surveillance service must be attentive, and the healthcare services should be sensitive to the notification of this disease, with professionals trained to identify suspected cases of the disease and to notify promptly to ensure early diagnosis and treatment, in addition to being active in epidemiological investigation and health education initiatives for the public [53]. The limitations of the present study were, among others, related to this cited misclassification bias, particularly if we consider that in almost half of the state municipalities (220; 44.2%), no leptospirosis cases were reported in any of the 13 evaluated years. However, even with the possible notification biases, the analysis of these data is extremely valuable for healthcare agencies, as it enables determining the trends of various health issues and thus directs efforts and resources to anticipate risk situations making surveillance and control more effective.

In the context of the one health concept, improving epidemiological surveillance methods is essential to control leptospirosis in Brazil, considering the impact already caused by this disease in the country. However, this improvement depends on the epidemiological characteristics of the disease, such as its spatial and temporal distribution and association between variables. These parameters are important because they help in understanding the health–disease process and in implementing educational, prophylactic, and targeted control measures and monitoring and surveillance practices. This study evaluated the spatial and temporal dynamics of leptospirosis in Rio Grande do Sul and showed the highest incidence regions and the highest infection risk years, thereby allowing a satisfactory disease incidence prediction. Through the use of GAM and SARIMA models, we present an approach that intends to help the understanding of disease occurrence in an endemic area in southern Brazil. Furthermore, the prediction model can be made more dynamic after including the current data; additionally, more complex predictive models can be developed by considering climatic, environmental, and socioeconomic variables for a more accurate prediction. Future studies could use the results obtained through these models to evaluate other time frames or smaller geographic areas, particularly those in which the disease is more frequent, for example, the Centro Oriental Rio Grandense and the Metropolitan of Porto Alegre mesoregions. It would be important to conduct studies that assess the causality of leptospirosis in these regions with the highest

occurrence, especially considering its occurrence profile, which is more linked to the urban or rural environment. This could also address the high underreporting observed in most municipalities, especially those located in the southern and western regions of Rio Grande do Sul. Therefore, we consider that the proposed model for leptospirosis incidence prediction could serve as a tool for surveillance, considering the growing need to clarify the dynamics of this neglected disease in Brazil.

## Supporting information

**S1 Fig.** Goodness-of-fit graphics of possible Generalized Additive Models(GAM) for Poisson, Gaussian and Negative Binomial models using: (a) thin plate regression; (b) duchon splines; (c) cubic regression spline; (d) b-spline; and (e) p-splines smoothness terms.
(TIF)

**S1 Table. Goodness-of-fit summary of possible Generalized Additive Models (GAM) using different Families and Smoothness terms***.** p<0,001; [1]Un-Biased Risk Estimator (UBRE), [2]Generalized Cross Validation and [3]Restricted Maximum Likelihood were used to identify appropriate smoothness.
(PDF)

**S2 Table. Goodness-of-fit summary of possible Seasonal Autoregressive Integrated Moving Average (SARIMA) models*.** Applied model; [1]AIC: Akaike's Information Criterion; [2]Mean absolute percentage error.
(PDF)

## Acknowledgments

We are grateful to the workersof Sistema Único de Saúdeof Rio Grande do Sul (SUS/RS) not only for their efforts to control and mitigate infectious diseases but also for providing the fundamental data for this analysis.

## Author Contributions

**Conceptualization:** Alessandra Jacomelli Teles, Bianca Conrad Bohm, Fábio Raphael Pascoti Bruhn.

**Data curation:** Bianca Conrad Bohm, Nádia Campos Pereira Bruhn, Fábio Raphael Pascoti Bruhn.

**Formal analysis:** Bianca Conrad Bohm, Nádia Campos Pereira Bruhn, Fábio Raphael Pascoti Bruhn.

**Funding acquisition:** Fábio Raphael Pascoti Bruhn.

**Investigation:** Alessandra Jacomelli Teles, Nádia Campos Pereira Bruhn, Fábio Raphael Pascoti Bruhn.

**Methodology:** Bianca Conrad Bohm, Nádia Campos Pereira Bruhn, Fábio Raphael Pascoti Bruhn.

**Project administration:** Alessandra Jacomelli Teles, Fábio Raphael Pascoti Bruhn.

**Resources:** Fábio Raphael Pascoti Bruhn.

**Software:** Bianca Conrad Bohm, Nádia Campos Pereira Bruhn, Fábio Raphael Pascoti Bruhn.

**Supervision:** Nádia Campos Pereira Bruhn, Fábio Raphael Pascoti Bruhn.

**Validation:** Bianca Conrad Bohm, Nádia Campos Pereira Bruhn, Fábio Raphael Pascoti Bruhn.

**Visualization:** Bianca Conrad Bohm, Suellen Caroline Matos Silva, Fábio Raphael Pascoti Bruhn.

**Writing – original draft:** Alessandra Jacomelli Teles, Fábio Raphael Pascoti Bruhn.

**Writing – review & editing:** Suellen Caroline Matos Silva, Nádia Campos Pereira Bruhn, Fábio Raphael Pascoti Bruhn.

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
