## [Decision Letter · Decision Letter 0]

24 Oct 2022

Dear Ms Silva,

Thank you very much for submitting your manuscript "Spatial and temporal dynamics of leptospirosis in South Brazil: a forecasting and non-linear regression analyses" for consideration at PLOS Neglected Tropical Diseases. As with all papers reviewed by the journal, your manuscript was reviewed by members of the editorial board and by several independent reviewers. In light of the reviews (below this email), we would like to invite the resubmission of a significantly-revised version that takes into account the reviewers' comments. 

Please make sure to address all comments and recommendations indicated by the reviewers.

We cannot make any decision about publication until we have seen the revised manuscript and your response to the reviewers' comments. Your revised manuscript is also likely to be sent to reviewers for further evaluation.

Sincerely,

Mabel Carabali, M.D., M.Sc., Ph.D.,

Academic Editor

Victoria Brookes

Section Editor

Please make sure to address all comments and recommendations indicated by the reviewers

Reviewer's Responses to Questions

**Key Review Criteria Required for Acceptance?**

**Methods**

-Are the objectives of the study clearly articulated with a clear testable hypothesis stated?

-Is the study design appropriate to address the stated objectives?

-Is the population clearly described and appropriate for the hypothesis being tested?

-Is the sample size sufficient to ensure adequate power to address the hypothesis being tested?

-Were correct statistical analysis used to support conclusions?

-Are there concerns about ethical or regulatory requirements being met?

Reviewer #1: (No Response)

Reviewer #2: - Objective requires a minor revision related to the resolution for spatial analysis and time-series and forecasting analysis.

- To obtain a stronger evidence, municipalities with enough data (Porto Alegre and couple more) could be analysed using GAM and SARIMA.

**Results**

-Does the analysis presented match the analysis plan?

-Are the results clearly and completely presented?

-Are the figures (Tables, Images) of sufficient quality for clarity?

Reviewer #1: (No Response)

Reviewer #2: - Figures quality requires revision.

- GAM and SARIMA over selected municipalities could contribute to the analysis

**Conclusions**

-Are the conclusions supported by the data presented?

-Are the limitations of analysis clearly described?

-Do the authors discuss how these data can be helpful to advance our understanding of the topic under study?

-Is public health relevance addressed?

Reviewer #1: (No Response)

Reviewer #2: - Adequate to the objective, even though time series analysis could be improved.

**Editorial and Data Presentation Modifications?**

Reviewer #1: (No Response)

Reviewer #2: (No Response)

**Summary and General Comments**

Reviewer #1: (No Response)

Reviewer #2: The authors analyse leptospirosis cases' spatial and temporal patterns in Rio Grande Do Sul state, South Brazil. They found that some cluster municipalities amassed most cases, and sanitary conditions could explain its incidence. Additionally, they found GAM appropriate to model the "trend" of time series and SARIMA for forecasting. 

I suggest applying GAM and SARIMA to analyse the "cluster municipalities" temporal behaviour of leptospirosis incidence. Other detailed comments for minor revision are in the attached document.

PLOS authors have the option to publish the peer review history of their article (what does this mean?). If published, this will include your full peer review and any attached files.

Reviewer #1: No

Reviewer #2: No
---

## [Decision Letter · Decision Letter 1]

26 Jan 2023

Dear Ms Silva,

Thank you very much for submitting your manuscript "Spatial and temporal dynamics of leptospirosis in South Brazil: A forecasting and  non linear regression analysis" for consideration at PLOS Neglected Tropical Diseases. As with all papers reviewed by the journal, your manuscript was reviewed by members of the editorial board and by several independent reviewers. The reviewers appreciated the attention to an important topic. Based on the reviews, we are likely to accept this manuscript for publication, providing that you modify the manuscript according to the review recommendations. 

Please refer to the reviewers comments and make sure that you address all the comments and recommendations, including formatting the figures and tables as recommended by the journal's guidelines.

Sincerely,

Mabel Carabali, M.D., M.Sc., Ph.D.,

Academic Editor

Victoria Brookes

Section Editor

Dear authors, thank you for providing a revised version of the manuscript.

Please refer to the reviewers comments and make sure that you address all the comments and recommendations. Including formatting the figures and tables as recommended by the journal's guidelines.

Reviewer's Responses to Questions

**Key Review Criteria Required for Acceptance?**

**Methods**

-Are the objectives of the study clearly articulated with a clear testable hypothesis stated?

-Is the study design appropriate to address the stated objectives?

-Is the population clearly described and appropriate for the hypothesis being tested?

-Is the sample size sufficient to ensure adequate power to address the hypothesis being tested?

-Were correct statistical analysis used to support conclusions?

-Are there concerns about ethical or regulatory requirements being met?

Reviewer #1: See attached PDF file

Reviewer #2: Meets the required items.

**Results**

-Does the analysis presented match the analysis plan?

-Are the results clearly and completely presented?

-Are the figures (Tables, Images) of sufficient quality for clarity?

Reviewer #1: Figures need more work since quality still suffer from pixelation when zooming in. See attached PDF file.

Reviewer #2: Meets the required items, check the quality of figures.

**Conclusions**

-Are the conclusions supported by the data presented?

-Are the limitations of analysis clearly described?

-Do the authors discuss how these data can be helpful to advance our understanding of the topic under study?

-Is public health relevance addressed?

Reviewer #1: See attached PDF file

Reviewer #2: Objctives are clearly answered.

**Editorial and Data Presentation Modifications?**

Reviewer #1: See attached PDF file

Reviewer #2: (No Response)

**Summary and General Comments**

Reviewer #1: See attached PDF file

Reviewer #2: (No Response)

PLOS authors have the option to publish the peer review history of their article (what does this mean?). If published, this will include your full peer review and any attached files.

Reviewer #1: No

Reviewer #2: No

Figure Files:

Data Requirements:

Reproducibility:

References

---

## [Editor Report · Decision Letter 2]

13 Mar 2023

Dear Ms Silva,

We are pleased to inform you that your manuscript 'Spatial and temporal dynamics of leptospirosis in South Brazil: A forecasting and  non linear regression analysis' has been provisionally accepted for publication in PLOS Neglected Tropical Diseases.

Best regards,

Mabel Carabali, M.D., M.Sc., Ph.D.,

Academic Editor

Victoria Brookes

Section Editor

Please address all minor editorial aspects mentioned by the reviewers.

---

## [Editor Report · Acceptance letter]

11 Apr 2023

Dear Ms Silva,

We are delighted to inform you that your manuscript, "Spatial and temporal dynamics of leptospirosis in South Brazil: A forecasting and  non linear regression analysis," has been formally accepted for publication in PLOS Neglected Tropical Diseases.

Best regards,

Shaden Kamhawi

co-Editor-in-Chief

Paul Brindley

co-Editor-in-Chief
